# Examining self-employment policies for persons with disabilities in South Africa: Perspectives from policy actors

Luther Lebogang Monareng[1,2]*, Shaheed Mogammad Soeker[3], Deshini Naidoo[1]

1 Department of Occupational Therapy, College of Health Sciences, University of KwaZulu-Natal, Durban, South Africa, 2 University College London, London, United Kingdom, 3 Occupational Therapy Department, University of the Western Cape in South Africa, Bellville, South Africa

* lebeganglolo@gmail.com, monarengl@ukzn.ac.za

## Abstract

### Background

Despite robust global and national efforts to promote inclusive development, a significant gap persists in countries such as South Africa's self-employment policies for persons with disabilities. The existing legislative framework, although well-intentioned, lacks clear and comprehensive guidance on self-employment as a viable placement option for persons with disabilities. Consequently, this ambiguity hinders effective policy implementation, limiting economic empowerment and social inclusion. This research aimed to explore the existence of self-employment-specific policies for persons with disabilities and policy actors' involved in South Africa.

### Methods

The participants (n=47) had an average of 10 years of experience in self-employment for persons with disabilities, holding qualifications ranging from no formal education to master's degrees. This qualitative study ensured transparent and systematic reporting using the Consolidated Criteria for Reporting Qualitative Research (COREQ) guidelines. Purposive and snowball sampling were utilised to recruit participants. Data were collected using a piloted question guide and analysed using the NVIVO software. Data was analysed thematically. Ethics clearance, relevant gatekeepers' permission and informed written consent from participants were obtained.

### Results

Two themes emerged, namely, theme one: The status quo on self-employment-specific policies for persons with disabilities. Participants reported on the absence of explicit policies on self-employment for persons with disabilities, the lack of effectiveness in inclusive South African legal frameworks and their lack of

**Data availability statement:** All raw data files are available from Open Science Framework (OSF). The full reference is as follows: Monareng LL. PhD Raw Data. Data collection from 2023 to 2024 [Internet]. OSF; 2025. Available from: https://osf.io/pdfks/.

**Funding:** This research was supported by i) South Africa's National Research Foundation (NRF), ii) University Capacity Development Grant Funding (UCDP) from the University of KwaZulu Natal and iii) The University of KwaZulu-Natal's Step Up Programme, Department of Higher Education and Training (DHET) and Newton Fund. The funders had no role in study design, data collection and analysis, decision to publish, or preparation of the manuscript.

**Competing interests:** The authors have declared that no competing interests exist.

impact on promoting self-employment opportunities. Theme two: policy actors' involvement in self-employment-specific policies for persons with disabilities. Participants reported on the roles and responsibilities of policy actors and strategies to promote self-employment opportunities for persons with disabilities through policy reforms.

## Conclusions

The research revealed a complex policy landscape where the absence of self-employment-specific policies for persons with disabilities coexists with potentially leverageable inclusive frameworks. These coexist with potential policy actors who have the potential to facilitate implementation. Leveraging existing policies through effective implementation and targeted policy reforms would ensure the full participation of persons with disabilities in self-employment. Key policy actors should familiarise themselves with the existing legal framework and emphasise enforcement and consequence management to ensure policies are implemented effectively. Furthermore, a coordinated approach is necessary, involving: a single or integrated system or database to streamline policy implementation and monitoring; a targeted approach that prioritises persons with disabilities in self-employment; and policies that allow for target setting, accurate measurement of targets, and effective monitoring and evaluation. Thus, a policy brief outlining the key findings should be considered, drafted, and submitted to the relevant government department (e.g., Department of Employment and Labour) for further action.

## 1. Background

The background section introduces the focus of this study, namely, self-employment-specific policies for persons with disabilities, using Walt's triangle model. This background established a foundation by integrating global and South African contexts, which are later referenced in the discussion section in relation to the study's findings. Specifically, the background is structured under the following main headings: Overview, Walt's triangle model, Global context, Local context and Study contribution.

### 1.1. Overview

Self-employment represent a preferred work placement for persons with disabilities [1,2], encompassing various activities, including production (e.g., furniture, clothing), service provision (e.g., hairdressing, cyber-cafe management), and trading (e.g., shop ownership) [3]. It is vital for persons with disabilities as it offers a flexible means to earn a living in diverse settings, whether urban or rural, formal or informal [3]. This type of employment can be pursued full-time or part-time, accommodating individual needs. However, research in this field remains sparse [4,5], i.e., not assisting towards lowering the global unemployment crisis. In response, this research unpacks the self-employment policy aspect using Walt et al's (1994) [6] triangle model, which can be applied retrospectively and prospectively. The triangle model incorporates four

interrelated components, namely the context, content and process on each side, and actors at the centre of the triangle. The context is the South African environment; however, international links are made to highlight the hierarchical structure of legal frameworks [7]. Content refers to self-employment-specific policies for persons with disabilities, while process focuses on how contextual (South African) policies are formed. Lastly, the actors or policy actors include policy informers, makers, implementors, and monitors.

## 1.2. Walt's triangle model - context, content, process and actors

The interrelatedness of Walt et al's (1994) [6] model's four components implies that one cannot discuss one element without the others. For instance, the environment (context), including politics, affect the processes followed when formulating policies (content) and outcomes thereafter. This suggests that a democratically elected political party runs a country's affairs, including leading the formulation and implementation of relevant policies through a lengthy consultative process with the country's citizens [8]. In other words, the content or policy outcomes need to appeal and address citizens' or context-specific concerns by following area-specific due processes. To make this document reader-friendly, some of Walt's model components are implicit, while others are explicitly mentioned.

Moreover, it is worth noting that this research will focus more on content (policy) and actors (those involved), which is in line with this study's focus, as elaborated below. Walt et al's (1994) [6] model emphasises that a comprehensive and enforceable content or legal framework is crucial. This links to facilitating success in self-employment and safeguarding persons with disabilities' fundamental rights, including well-being, dignity, community participation and adult work engagement [3,9]. Frug et al. (2006) [7] state that the legal framework is typically hierarchical, with global legislation informing national policies. When there is strategic alignment, global legislation informs national planning, implementation, and evaluation, which provides valuable guidance to nations [7]. Thus, with suitable actors, one would assume that content (policy) implementation should be feasible towards promoting persons with disabilities. Target setting, enforcement and ongoing monitoring will be paramount. These imply that a transparent multi-level system from the macro (legislation) to the micro or basic (end user) level is vital [10–13].

## 1.3. Global level (context) - laws and institutions on self-employment-specific policies for persons with disabilities

Although initial research by the authors of this article indicated limited international literature and a paucity of literature specific to the content on self-employment for persons with disabilities [5], at the global level (context), key institutions or actors drive inclusive development through pivotal initiatives, promoting a culture of inclusivity and equal opportunities. These are equally important in self-employment, given its inherent competitive nature. The United Nations' 2030 Agenda for Sustainable Development Goals (SDGs) [9], the World Health Organization's (WHO) Community-Based Rehabilitation (CBR) Matrix [14], and the International Labour Organization's (ILO) [3] disability inclusion strategies are cornerstones of global efforts to promote inclusive development. These global initiatives are reinforced by legislation and frameworks (content), which underpin and support self-employment for persons with disabilities. Specifically, the Declaration of Alma-Ata (1978) [15], the SDG plan, and the CBR framework provide the necessary foundation and guidelines to advance the needs of persons with disabilities. Moreover, the Alma-Ata Declaration (1978) [15] represented a pivotal moment in the global health landscape, prioritising the health needs of vulnerable populations, including children, women, and persons with disabilities, to address pervasive global health inequalities. The declaration emphasises the importance of optimal health as a multifaceted concept, encompassing physical well-being and social determinants such as shelter, education and employment, which collectively foster active community participation and economic engagement.

Similarly, the United Nations Development Programme (UNDP) is crucial in advancing sustainable development and fostering a more equitable and just world, especially for disadvantaged nations. This commitment is reflected in the United

Nations' 17 SDGs or global goals, which are aimed to be achieved by 2030, prioritise the improvement of the lives of vulnerable populations, including children, women, and persons with disabilities, through a multifaceted approach [9]. CBR [14] complements the abovementioned policies, a pivotal framework developed by the WHO in the 1980s. CBR's primary objective is to enhance the quality of life for persons with disabilities through holistic and inclusive initiatives, such as skills training, promoting independence, and empowerment. Community-based rehabilitation's underlying principles, inclusion, participation, empowerment, and self-advocacy, align with goals on determinants of health, which are essential for one's well-being [14]. The CBR matrix comprises five interconnected domains: health, education, social, empowerment, and livelihood, each with five sub-domains that provide a framework for implementation. Notably, the livelihood domain encompasses critical components of economic empowerment, including skills development, wage employment, financial services, social protection, and self-employment [14].

## 1.4. Local level (context) - laws and institutions on self-employment-specific policies for persons with disabilities

Informed by global laws, at the local or national level (context), South Africa's legislative framework (implicit or inclusive policies) provides a solid generic foundation for promoting disability rights and inclusion and should be leveraged. Key documents guiding disability rights and inclusion in this country are guided by the Constitution of the Republic of South Africa, 1996, which enshrines and protects the rights of persons with disabilities [8]. For instance, the South African White Paper on the Rights of Persons with Disabilities (2016) [16] highlights explicitly the significance of economic empowerment in promoting inclusive development and equal opportunities. Specifically, the document emphasises the fundamental right to work and employment for persons with disabilities, advocating for non-discrimination, inclusive employment opportunities and self-employment initiatives. By promoting these initiatives, the White Paper on the Rights of Persons with Disabilities aims to empower persons with disabilities to start their businesses, fostering entrepreneurship and financial independence [16].

Although generic, South African laws collectively foster an inclusive environment, promote social transformation, and aim to eradicate poverty by ensuring equal opportunities for persons with disabilities. See the discussion section for more details on similar laws. These are good-intentioned content (legislation) in both global and local contexts, yet efficient implementation and impact are yet to be seen. One would then argue that an explicit, action-driven and enforceable approach is of the essence. This is supported by a critique by authors such as Grobbelaar-du Plessis and Njau (2019) [17] on the lack of policy implementation in the South African context. Their critique can further be applied to various government departments crucial in implementing the country's legal framework. These include the Department of Planning, Monitoring and Evaluation (DPME), which oversees the implementation of national policies and programs, and the Department of Small Business Development (DSBD), which promotes entrepreneurship and supports small businesses.

Additionally, the Department of Women, Youth and Persons with Disabilities (DWYPD) prioritises empowering and including vulnerable groups. At the same time, the Department of Trade, Industry and Competition (thedtic) drives economic growth and development through trade, industry, and competition policies. These laws and institutions can be leveraged to ensure the mainstreaming of persons with disabilities' daily lives.

## 1.5. Generic law-making process in South Africa

As discussed below, the generic and lengthy law-making process in South Africa is for consideration when suggesting policy reform in self-employment-specific policies for persons with disabilities. More importantly, including persons with disabilities in the process should be prioritised. For the laws and institutions outlined above to function effectively and align with international standards, the South African government relies on processes known as the three arms of government [18], which provide a system of checks and balances: namely the legislature (parliament), which formulates and enacts laws and ensures government accountability, the executive (cabinet), which implements and governs the country, and the judiciary (courts), which interprets and upholds the law. This separation of powers is essential in maintaining

the independence and distinctness of each branch or arm of government [18]. Although this doctrine provides clarity, its application can be nuanced, and the lines between branches' functions can sometimes become blurred [18]. Below is an overview of the three arms of government and their processes to provide context.

The parliament, which makes the law, comprises the National Assembly and the National Council of Provinces. A draft or proposal is introduced at the parliamentary level, where it is debated. If approved, it progresses to the next stage as a bill and is then sent out for public comments. These comments are subsequently debated or discussed in parliament and incorporated. Finally, the document is signed into law by the government (president), becoming a binding statute [18]. Refer to Fig 1.

Building on, understanding the following key legal terms is essential for navigating the South African legal framework [18]:

• *The Constitution*: It serves as the country's supreme law within defined limitations.

• *An Act*: Represents national legislation that expands on the Constitution, such as the Labour Relations Act (LRA), which originates from Section 23 of the Constitution, addressing employment rights.

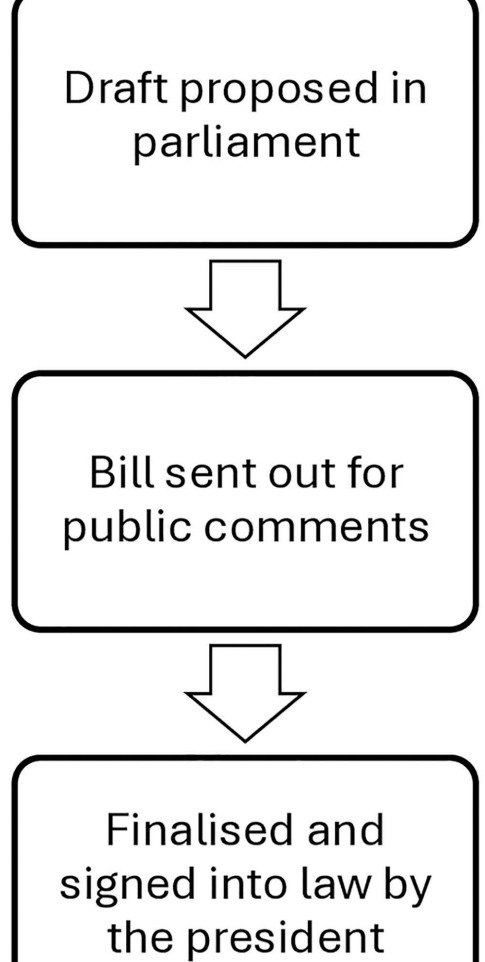

**Fig 1. Law-making process in South Africa.**

- *Policy*: Refers to guidelines limited to a particular department or company, such as the Department of Employment and Labour.

- *Gazette*: It is an official government publication platform that disseminates information on new laws, for instance. It is more like a government newsletter or newspaper.

- *Proclamations*: Are official announcements or declarations.

The judiciary, or courts, plays a crucial role in interpreting and clarifying the law in cases where disputes arise. It comprises the Constitutional Court, the Supreme Court of Appeal, and High Courts [18]. For instance, legal litigation or grievances related to the National Health Insurance (NHI) Act [19] or Compensation for Occupational Injuries and Diseases (COID) Acts are adjudicated within these courts when necessary. Access to the judiciary is, however, associated with costs and is believed to favour those with financial means and disadvantage the majority of South Africans, given the country's legacy of systemic oppressive government [20].

The executive, as the implementor of the law, is responsible for executing policies. The executive involves the government, supported by ministries, such as the Health or Labour Ministry, respectively, which deliver services to citizens. For example, the Labour Ministry (Department of Employment and Labour) is implementing the amended COID Act. For effective and impactful service delivery, it is crucial to address the ingrained culture of self-interest (crime) that some officials perpetuate [21,22]. Furthermore, the Department of Employment and Labour has Labour inspectors who monitor and enforce compliance, issue fines or escalate non-compliance matters to appropriate authorities such as courts [18]—actors in this research query the inspectors' effectiveness and the punitive approach to non-compliance. Alternative dispute resolution channels in this context include the Commission for Conciliation, Mediation and Arbitration (CCMA), which, among other duties, ensures employers pay employees any owed wages.

## 1.6. Study contribution

Despite the above robust global and national efforts to promote inclusive development, there seems to be a significant gap that persists in South Africa's self-employment policies for persons with disabilities. Although well-intentioned, the existing implicit legislative framework lacks clear and comprehensive guidance on self-employment as a viable placement option for persons with disabilities. Consequently, this ambiguity hinders effective policy implementation, limiting economic empowerment and social inclusion opportunities as intended by the legislative framework. As such, this study addressed the following research question: What self-employment-specific policies exist for persons with disabilities in South Africa? In alignment with the question, the study explored two objectives as informed by South African policy actors: i) The existence of self-employment-specific policies for persons with disabilities and ii) Policy actors' involvement in the development and implementation of self-employment-specific policies for persons with disabilities. Overall, this study offers valuable insights for those involved in this field. The approach can be adapted and replicated in other countries to strengthen self-employment-specific policies for persons.

## 2. Methods

Following the Consolidated Criteria for Reporting Qualitative Research (COREQ) guidelines [23], this qualitative study ensured transparent and systematic reporting. The COREQ 32-item checklist for interviews and focus groups informed the reporting structure, organised under three key domains: researcher positionality, study design, and data analysis and findings [23].

*Research team and reflexivity:* This research was conducted by three researchers, comprising the two co-authors and the corresponding author. Author disclosures, including qualifications and affiliations, are provided on the title page to promote transparency and accountability. All the authors have experience in conducting qualitative studies. Pre-interview briefings facilitated open communication, allowing participants to query the researcher's background, objectives, and

methods. Data analysis remained grounded in empirical evidence to maintain methodological rigour, minimising potential biases and assumptions. A strategic approach combining purposive sampling and snowballing techniques was employed to optimise participant diversity and richness [24]. Multichannel recruitment strategies were utilised to approach potential participants by incorporating telephone calls and emails. In-depth, semi-structured in-person and online interviews were conducted with n = 47 participants across South Africa. Participants (representatives) were selected based on the inclusion criteria in Table 1 below. Those who did not meet the criteria were excluded.

The data collection duration was from 01 June 2023–15 November 2024. The research question, objectives, and literature review guided the construction of a comprehensive question guide and probing questions. A preliminary validation phase involved pilot-testing the guide with four eligible individuals, ensuring conceptual alignment and ensuring there was no ambiguity. Iterative refinements followed, incorporating terminological clarifications (e.g., using synonyms such as 'obstacles' alongside 'barriers'). Interviews ranged from online to face-to-face, individual to triad interviews. English was the primary language used, except with persons with disabilities, where their preferred local language (isiZulu) was utilised. On the interview day, participants received detailed information about the research and signed the consent form. Each interview was audio-recorded and transcribed verbatim, ensuring rigour in data capturing. Participants were allowed to reflect, revise, and expand their responses during the interview. Data collection was stopped when thematic data saturation was attained, indicating sufficient data had been collected [24,26].

Data organisation and coding were done using NVIVO software. A three-member analytical team was involved, consisting of the corresponding author and two co-authors, who collaborated on the coding process. An integrative thematic analysis, blending inductive and deductive reasoning [26], revealed two overarching themes and associated sub-themes substantiated by participant narratives, i.e., quotes. There was a balance between inductive codes (emerging from the data and quotes) and deductive coding (informed by the research focus and question guide). This hybrid approach ensured methodological integrity [27], credibility, and trustworthiness [28]. As outlined previously, transparent research practices were maintained by applying strict inclusion criteria [26]. To enhance analytical validity, the authors conducted initial data interpretations, which were then peer-reviewed and critiqued with the team [26,29].

Ethical clearance was formally granted by the University of KwaZulu-Natal's Biomedical Research Ethics Committee (BREC) BREC/00004655/2022. Permission or access approvals were additionally secured from relevant gatekeepers, and participants were fully informed about the study through detailed information sheets and provided signed informed

Table 1. Policy actors' inclusion criteria.

| Policy actor | Inclusion criteria |
| --- | --- |
| Persons with disabilities | - Meet the 18–65 years of the South African working age<br>- Have been running a microenterprise for at least three years and making a living from it [25]<br>- Have a disability |
| Occupational therapists | - Registered with their professional body, the Health Professions Council of South Africa (HPCSA) and actively practicing as an occupational therapist (at Learners with Special Education Needs School (LSEN), Clinician or academia)<br>- Have three or more years of work experience and<br>- Know about vocational rehabilitation |
| Persons with disabilities organisations | - Working for at least three years<br>- With knowledge about self-employment in microenterprises for persons with disabilities |
| Government departments included DoE, DWYPD and thedtic. | - Working for at least three years<br>- With knowledge about self-employment in microenterprises for persons with disabilities |

consent before participating. Upholding research ethics, this study strictly adhered to foundational principles, prioritising participant privacy, confidentiality, and anonymity to protect participants' rights and dignity.

## 3. Findings

Although not explicitly mentioned, the findings below are presented in alignment with Walt et al's (1994) [6] model with more emphasis on content (policy) and actors (those involved).

### 3.1. Demographics

There was a total of n = 47 participants (actors or policy actors) in this research. All of them met the predefined inclusion or eligibility criteria. Refer to Table 2 below for more details on the demographics of the policy actors. Their knowledge and exposure to self-employment for persons with disabilities averaged 10 years of experience. Their qualifications ranged from having no formal education to having a master's degree.

### 3.2. Themes

Two themes emerged from the data, theme one: The status quo on self-employment-specific policies for persons with disabilities and theme two: policy actors' involvement in self-employment-specific policies for persons with disabilities. Refer to Table 3 below for the themes and subthemes.

**3.2.1. The status quo on self-employment-specific policies for persons with disabilities.** This theme focuses on the status quo of self-employment-specific policies for persons with disabilities in South Africa, with three corresponding subthemes. The subthemes delve into the following key issues: the absence of explicit policies on self-employment for persons with disabilities, the effectiveness of South African inclusive legal frameworks, and their impact on promoting self-employment opportunities.

3.2.1.1. The lack of explicit policies on self-employment for persons with disabilities: The policy actors reported their lack of awareness regarding specific policies supporting self-employment for persons with disabilities. Furthermore, they noted that existing policies narrowly focus on self-employment opportunities for this population. The participants had the following to say:

**Table 2. Demographics of policy actors.**

| Pseudonyms | n | Average years of experience | Policy actor (Participants) | Institution/setting | Location | Qualifications range |
|---|---|---|---|---|---|---|
| BN, CK, GM, HS, JM, MB, PM, SN, TM, VM, | 10 | 05 | Persons with disabilities | Microenterprise owners | KwaZulu-Natal province | No formal education to <Grade 12 |
| AP, Arthur, Blue One, Blue Flower, Brown, Butterfly, Emerald, Hedgehog, JR, Lune, Mac, Participant, Participant 00, Participant 05, Number 5, Shera, Taxido Cat | 17 | 14 | Occupational therapist | Academics, clinicians, and LSEN School-based | South Africa | Bachelor's to master's degree |
| Bear, Blue, Bond, MB2, MM, Vision | 06 | 14 | Persons with disabilities organisation | Non-profit organisations and private | South Africa | Training on the job to master's degrees |
| Ginger, Green, Grey, LD, Lion, Mzamusa, Ntoborozi, Pinky, Purple, Rakgadi, SP, Sugalite, Tau, Tebogo | 14 | 08 | Government officials | DoEL, DWYPD & thedtic) | South Africa | Diploma to master's degree |
| **Total** | 47 | 10 | – | | | |

**Table 3. Findings themes and corresponding subthemes.**

| Themes | Subthemes |
|---|---|
| 3.2.1 The status quo on self-employment-specific policies for persons with disabilities | 3.2.1.1. The lack of explicit policies on self-employment for persons with disabilities |
| | 3.2.1.2. Inclusive legal frameworks supporting persons with disabilities in South Africa |
| | 3.2.1.3. Impact of inclusive legal frameworks supporting persons with disabilities in South Africa |
| 3.2.2. Policy actors' involvement in self-employment-specific policies for persons with disabilities | 3.2.2.1. Policy actors and their role in policy making |
| | 3.2.2.2. A way forward and promoting self-employment through policy reforms for persons with disabilities |

*"Current policies are not targeted specifically at persons with disabilities. They deal holistically with designated groups. So, persons with disabilities are just, by the way... The priorities then go to youth, and they go to gender, which is women."* (SP, Government official)

*"I'm not aware that there is a specific policy for persons with disabilities when it comes to entrepreneurship."* (Sugalite, Government official)

The participants emphasised that the policies they are aware of were not explicitly targeted at persons with disabilities but rather inclusive in a broad sense, without prioritising persons with disabilities and their needs. Moreover, they expressed concerns that existing policies disproportionately favour able-bodied individuals and prioritise employment over self-employment for persons with disabilities. The participants shared the following:

*"What I know about South Africa is that a whole lot of Acts out there that actually advocate for employment of people with disabilities, and usually the focus is for them to be employed rather than them being self-employed... But I am yet to see the coverage of those who require to be self-employed."* (Ginger, Government official)

*"So the policy in South Africa sometimes they give us big challenge because the policies are no including the disabled people... They write the policy, they think about the abled people only."* (MB, Persons with disabilities)

3.2.1.2. Inclusive legal frameworks supporting persons with disabilities in South Africa: Participants reported that South African law and legal frameworks are inclusive, covering the general population and citizens in the open labour sector while providing support and accommodations for persons with disabilities. They believe that persons with disabilities ate not the primary target. The participants shared the following:

*"You know the reality of South Africa from a legislative perspective has got one of the most progressive and inclusive legislations in the world."* (Bond, Persons with disabilities organisation representation)

*"Even though I don't understand them* [legal frameworks/policies] *well, but I know that a disabled person has the right to establish their own business. They also have a right to get funding."* (Vision, Persons with disabilities organisation representation)

*"Everyone in South Africa is encouraged to be capacitated* [trained] *in terms of the Constitution. Everyone should be capacitated* [including] *persons living with disabilities, remember they are the most disadvantaged."* (Ntomborozi, Government official)

The participants added that these legal frameworks enable and are sensitive to the needs of persons with disabilities. Some of the leading legal frameworks they mentioned include, but are not limited to, The Constitution of the Republic of South Africa, The White Paper on the Rights of Persons with Disabilities and the National Development Plan policy. The following are supporting quotes from the participants:

> *"I'm aware of the Constitution and also the Employment Equity Act… In terms of the Constitution of South Africa, every disabled person has a right to dignity and to be employed without being discriminated. Everybody has equal rights besides their disability."* (Green, Government official)

> *"The COID* [Compensation for Occupational Injuries and Diseases Act] *will cover them* [persons with disabilities]*…"* (Ginger, Government official)

> *"So, I think we do have the correct tools within the* [South African] *country. You've got your PEPUDA* [Promotion of Equality and Prevention of Unfair Discrimination Act]*, we've got your skills development at your EEA* [Employment Equity Act]*. We've got triple BEE* [Broad-Based Black Economic Empowerment (B-BBEE) policy] *and other prescripts."* (Tebogo, Government official)

> *"The policies that would really be relevant would be the procurement bill* [Preferential Procurement Policy Framework] *that I've just touched on. They say 7% of total procurement must be given to persons with disabilities."* (SP, Government official)

> *"If maybe there is a tender… If you are disabled, you stand a good chance to get that tender, if your things* [documentation] *are in order. So when you compete automatically it gives you that opportunity to be considered... So you already have advantage before those other people who are able* [bodied]*."* (Green, Government official)

> *"*[There are] *policies of inclusion and within the country* [South Africa]*, we've got a, a great, overarching policy which is the National Development Plan Policy that is having targets on persons with disabilities, either through skills development, employment as well as procurement. "* (Tebogo, Government official)

Additionally, participants noted the importance of supporting institutions and organisations, such as skills development entities, including Sheltered Employment Enterprises (SEE) and Sector Education and Training Authorities (SETAs), as they align with the legal frameworks. They view these institutions as crucial contributors to one being self-employed. Below are supporting quotes from the participants:

> *"They* [persons with disabilities] *are involved in SEE, Sheltered Employment Enterprises...."* (Ginger, Government official)

> *"When it comes to training, you have SETAs. They are actually looking to train persons with disability everywhere. So, as much as it's not self-employment, it is an enabler towards self-employment"* (Tau, Government official)

3.2.1.3. Impact of inclusive legal frameworks supporting persons with disabilities in South Africa: The participants noted that despite the government's promises and South Africa's implicit policies on self-employment for persons with disabilities, they remain ineffective in practice. They questioned the status quo and the impact of the above legal frameworks and entities, citing unattended basic human needs such as access to transport. Below are supporting quotations from the participants:

> *"Oh! I don't think that they* [policies] *have impact. If there is impact, that impact is minimal."* (Ntomborozi, Government official)

*"The theory* [polcies in writing]*, it's amazing. In practice, it's absolutely shocking."* (Bond, Persons with disabilities organisation representation)

*"Yes, the policies are there, and they help, but they are inadequate... I say they are inadequate because there are many people that are disabled but there are few that get assistance. So, it means that there is a lot that still needs to be done."* (Vision, Persons with disabilities organisation representation)

*"I'm gonna be frank now. On paper, those policies look fantastic… In football, a match is never won on paper. It's about executing the strategy that makes the team superior, regardless of what is on paper... The basic needs of persons with disabilities are not met... Transport is a huge issue... So yeah, execution from government* [is] *poor or non-existent.."* (Bear, Persons with disabilities organisation representation)

They expressed concerns about the inadequate implementation (execution) and enforcement (monitoring and evaluation) of these policies by relevant companies or institutions. The participants cited the lack of systems to measure policies' effectiveness and track progress towards targets and outcomes. As such, they added that policies do not translate into tangible benefits to address the needs of persons with disabilities but rather excuses from these relevant institutions. The participants shared the following:

*"I think the weakness in the system is consequence management... There's a number of departments that have not reached their targets. I think the department took an approach of naming departments that are not providing perfor-mance reports. Uh, in my experience it doesn't help to name and shame anybody. For the Employment Equity Act, it says 'fines will be imposed', so employment and Labour* [DoEL] *is not telling us if they've collected fines from anybody or what they're doing with that. If they had collected."* (Sugalite, Government official)

*"If you look at* [for instance] *where I work, I always ask supply chain, 'How much? How many businesses that are run by persons with disabilities, have we allocated business to?' They'll always come up with excuses. Like 'No, they don't declare their disability'. So, the execution or the implementation is problematic at the moment."* (SP, Government official)

*"The issues of monitoring and ensuring that everything has been done correctly, to make sure that those policies* [are implemented]*, we do not have that yet."* (Tau, Government official)

Additionally, they reported that barriers include governmental inefficiencies such as delayed processes, inadequate communication and suspected malpractice, which further erode trust in the government's ability to deliver on its policy promises. According to the participants, end users with means and access are compelled to seek alternative solutions, preferring to work with private initiatives or other sectors that can provide more effective support. Below are supporting quotes from the participants:

*"There will be a policy that will be made with good intentions, but when it comes to procurement officials, because they've got all selfish ambitions, they will give you all sorts of reasons why it cannot be implemented.."* (SP, Government official)

*"*[Department of] *Social Development… They are* [meant] *to support us* [persons with disabilities or their organisations]. *They don't support us... They don't care or they don't come to you to ask* [consult]*what you need so that you can reach your goal."* (MM, Persons with disabilities organisation representation)

**3.2.2. Policy actors' involvement in self-employment-specific policies for persons with disabilities.** Two subthemes emerged, one with a primary focus on the roles and responsibilities of policymakers and the other with a focus on strategies to promote self-employment opportunities for persons with disabilities through policy reforms.

3.2.2.1. Policy actors and their role in policy making: Participants reported that, although not performed by an individual or a single department, policy drafting and amendments primarily occurs within government departments. They emphasised that a multidisciplinary approach should involve collaboration among key role players or the various government departments including Department of Basic Education, Department of Transport and Department of Social Development. While participants noted that no single section is responsible for policy drafting, input is solicited through a consultative process with coordination and oversight provided by high-level or national authorities before being disseminated to the public for comments. The quotes that follow are subdivided into direct and indirect involvement in policy making to indicate the nature of roles played by different policy actors. The participants shared the following:

Direct involvement in policy making

*"In terms of policy development... It's not necessarily one person or one organization outside of government that develops policies... All departments at national level are policymakers...* [For instance] *I'm at a national level, so* [our focus] *is mainly looking at policy development, policy implementation where the coordination mechanism within government that looks at mainstreaming of persons with disabilities and advocating for access and participation at all levels across all rights... We do draft, I mean, for example, we were also involved in the drafting of the Convention on the Rights of Persons* [with disabilities] *and the White Paper itself... But further to that, as new policies, got public comment or legislation that we're aware of, we* [her section] *will ensure that there's disability mainstreaming in that."* (Sugalite, Government official)

*"At the at the national level, we get invitations and we have input... We have a multidisciplinary* [team]*."* (Tau, Government official)

*"We do the drafting* [of policy]. *We first do business case studies, see if there's a need for a regulation, then we do the regulation. We draft it. Once it's drafted and it gets signed off by the Minister, then it needs to be implemented and enforced so our inspectors enforces the regulations and at the Head Office."* (Purple, Government official)

Indirect (and limited) involvement in policy making

*"So I get only involved in policies that sort of include my role... Then I would be roped in."* (LD, Government official)

*"We* [her section at a government department] *don't do any policies for persons with disabilities, but we've got a unit that deals with employment equity issues."* (Mzamusa, Government official)

*"We* [his section at a government department] *implement what is out there, and then we advise if there's policy inadequacies... For examples... We give opinion to say 'Based on what's happening on the ground, this is what needs to happen' and then they* [responsible government departments] *are the ones that will then influence the Presidency and other Departments."* (SP, Government official)

3.2.2.2. A way forward and promoting self-employment through policy reforms for persons with disabilities: As a way forward, participants suggested that key role players in self-employment should leverage the existing legislative frameworks, familiarise themselves with them and develop them to favour persons with disabilities. They believe this will create a supportive environment that addresses their needs. Participants emphasised the need for impact-driven initiatives that prioritise persons with disabilities rather than grouping them with other marginalised groups, such as women and youth, which often take precedence. They advocated for a more robust approach, including integrated systems and affirmative action through legislation, where opportunities are specifically for persons with disabilities. Ultimately, participants called for a multifaceted approach, emphasising the need for accountability through enforcement mechanisms, consequence management and ongoing monitoring. In addition, the participants suggested that persons with disabilities could be more organised and engaged in policy-making. Below are some supporting quotes from the participants:

*"There's a lot of talk about youth participation. But what about persons with disabilities? Very less. So we need a more targeted approach specifically for persons with disability... We need targeted policies, targeted interventions... Measurable policies that are directed strictly to persons with disabilities. Then, we can measure the impact. So that we are not reactive... If you enforce and you start implementing policies, then it becomes workable, because you can then make people accountable... Hold stakeholders accountable, professionals and everybody else... Within Labour* [DoEL, the responsible personnel will be the] *inspectorate..."* (SP, Government official)

*"*[For a solution] *we need to have one platform... Try and sit under one roof and come up with one solution to say how do we integrate and move forward."* (Tau, Government official)

*"A number of the institutions say that it's difficult to find people with disabilities. They believe that if we have a database of, Uh, persons with disabilities who are in a state of readiness or, you know, or even know where they are and how to contact them. They would be able to achieve their targets."* (Sugalite, Government official)

*"Again, my plea is the execution part, the implementation part. If the policy is there already, then there must be serious monitoring on the implementation part to make sure that it really benefits the deserving targeted individuals."* (SP, Government official)

*"They* [persons with disabilities] *should become part* [active participants] *of the policymaking* [process]*, there is a slogan which says 'Nothing about us, without us', meaning that 'You cannot decide for me without me.'* [Persons with disabilities should] *take leadership positions, especially in government departments."* (Ntomborozi, Government official)

## 4. Discussion

Findings in this study highlighted the gaps between policy formulation and practical implementation, providing valuable insight for policymakers on economic inclusion for persons with disabilities. In line with the focus of this research, the policy actors' main input encompasses policy gaps and challenges, existing implicit frameworks and potential pathways to effective policy reform. Furthermore, aligning with Walt et al's (1994) [6] triangle model, the discussion below focuses on the following key points:

• Lack of or need for mainstreaming and integration of persons with disabilities into all aspects of society, including necessary skills training.

• Integrative and collaborative work among policy actors.

• Streamlining of efforts, including ensuring systems in implementation, accountability and ongoing monitoring for follow-through.

The findings reveal a concerning absence of self-employment-specific policies or content for persons with disabilities [6]. This is detrimental as it undermines the intentions of both global and local legislature and institutions seeking inclusivity through determinants of health [15]. The lack of targeted policies perpetuates the systemic exclusion of persons with disabilities from meaningful economic participation, social integration and overall well-being [3,9,14,30]. A prospective solution would be having clear, targeted policies and practical procedures for promoting self-employment as a viable placement option for persons with disabilities.

Despite this overt gap, the actors [6] are adamant that existing implicit or inclusive policies can be leveraged to ensure persons with disabilities' full participation in self-employment in a South African context. Effective leveraging can be achieved by ensuring implementation and consequence management. Although lacking specifics and deemed to favour abled-bodied individuals, such policies encompass the White Paper on the Rights of Persons with Disabilities [16], National Development Plan (NDP) [31], the South African Constitution [8], The Preferential Procurement

Policy Framework Act (PPPFA) of 2000 [32] and The Promotion of Equality and Prevention of Unfair Discrimination Act (PEPUDA) of 2000 [33]. Ginger, a policy actor, critiqued the South African legal frameworks, stating that it is biased towards employment over self-employment. To expand on these South African inclusive legal frameworks, consider the following points:

- Informed by the Constitution, which safeguards persons with disabilities' rights [8], the White Paper states inclusion goals. However, results are lacking due to various factors, including a lack of specificity in the policy and a lack of policy implementation. Championing self-employment support should focus on actionable steps.

- Drawing from section 9 of the Constitution [8], the NDP [31] policy aims to enhance skills, supported by SETAs, training persons with disabilities widely. Yet, it lacks focus on self-employment, neglecting post-training follow-through, which should be the focus. Tebogo, a policy actor, noted failures in meeting skills and placement targets.

- PPPFA is explicit about procuring goods and services from qualifying persons with disabilities in government tenders [32], but the implementation and follow-up remain weak. Among other factors, this could be attributed to the crime (corruption) committed by some officials. Herrington et al. (2010) [21] and Monareng et al. (2025) [22] argue that there should be zero tolerance for crime – those involved must face stringent legal consequences through the judiciary.

- In addition, PEPUDA [33] lacks enforcement, limiting persons with disabilities' economic inclusion. Ongoing monitoring is essential for follow-through.

Grobbelaar-du Plessis and Njau (2019) [17] corroborate on the South African government's lack of policy implementation culture, which stems from inadequate monitoring and evaluation, affecting the translation of legislation into practice. This undermines the lengthy processes associated with macro-level policy setting. Furthermore, the study highlights the absence of proper documentation and effective tracking regarding the inclusion of persons with disabilities into the South African economy, posing a significant challenge in addressing mainstreaming and their economic exclusion. On the other hand, other targets, such as providing opportunities for skills training to persons with disabilities, have not been met, further exacerbating their systemic exclusion [34]. Policy actors should collaborate and integrate their systems by considering an accessible, single database for persons with disabilities, which should make self-employment tracking a reality.

Overall, the existing implicit policies (content) collectively provide a foundation for promoting inclusivity and economic empowerment for persons with disabilities. Still, structural and systemic barriers hinder their implementation, including internal inefficiencies, lack of accountability, and non-compliance with policies. Additionally, there is a lack of targets and indicators, which makes it impossible to measure the impact of such inclusive policies. SP, a policy actor, alluded to officials' *"selfish ambitions"* (crime or corruption), adding to these challenges. This is supported by research [21,22]. Essentially, these challenges lead to long-term negative impact, including mistrust in actors or those involved in service delivery that could benefit persons with disabilities.

An exemplary global response to the COVID-19 pandemic demonstrated that effective outcomes depended on a proactive stance and ongoing communication strategies. The constant dissemination of concise messages, including promoting social distancing, successfully mainstreamed public health measures [35]. Similarly, the South African Revenue Service (SARS) or tax collection system [36], despite having challenges, has shown results through persistent follow-ups by their officials on non-compliant taxpayers. The above argument suggests that a comparable urgency and vigour, characterised by targeted communication and robust follow-up mechanisms, could be utilised and leveraged to address the plight of persons with disabilities. Replicating these approaches, tailored to persons with disabilities' needs, may offer a viable pathway to enhance their economic inclusion.

Moreover, inefficiencies in policy implementation are also observed in the highest laws, including the South African Constitution, which is known to be among the best in the world but lacks efficient implementation and meaningful impact

[37]. Additionally, to promote self-employment as a viable placement option for persons with disabilities, policy actors must acquaint themselves with the existing legal framework, especially in their line of work. Policy actors could be trained to mitigate the lack of policy implementation [17]. Effectiveness should be supplemented by enforcement and consequence management, as emphasised by the policy actors in this research.

In addition to targeted approaches, the actors cautioned against adding new tools, such as new policies, without implementation. They are adamant about leveraging existing laws and policy processes in South Africa [18]. According to one policy actor, SP, to achieve meaningful outcomes, interventions must be tailored to address the needs of persons with disabilities specifically. Even though law formulation is a complex and lengthy multidisciplinary process [6,18], it is still crucial to include key policy actors, such as persons with disabilities, in line with their slogan recited by a policy actor in this research, i.e., *'Nothing about us, without us'*. Among other strategies, including persons with disabilities could entail mainstreaming and addressing their expressed needs rather than mere lip service [38–40]. Targeted policies that are impact-driven should foster target setting, accurate measurement of targets, and effective monitoring and evaluation. For instance, regular site or field visits encompassing strict record reviews and offering needed support should be adhered to, as it is essential for accountability. As Walt et al. (1994) [6] supported, the overall focus should be on the interrelated context, content, process, and actors for a holistic and practical policy reform. This implies that the three arms of the South African government [18] should be leveraged, which is currently not the case.

Despite its inclusive policies and efforts to champion international policies, like the United Nations Convention on the Rights of Persons with Disabilities (UNCRPD), South Africa, like many other African countries, has not made significant progress in improving the lives of persons with disabilities. The citizens of this country continue to struggle to access basic needs, let alone specialised services such as inclusive employment opportunities or being self-employed [41]. One policy actor summed it up by stating that basic needs like transport should be attended to first. This stagnation critiques the assertion by Frug et al. (2006) [7] that legal frameworks hierarchically filter from global to local, as these frameworks remain unleveraged, leaving some policy actors in this research unsure about the reasons. The lack of use or leveraging likely stems from authorities' unwillingness, low priority, disinterest, absence of agency and attitudes that also manifest as inadequate resource allocation, including insufficient yearly budgets to support necessary initiatives. Such budgetary curtailments, while not exclusive to persons with disabilities, reflect broader economic struggles in South Africa, including cost curtailment and reducing staff in government jobs [42,43]. To address this, there should be dedicated teams or sections within existing offices to monitor and focus on executing existing policies alongside portfolios that employ committed actors willing to drive progress.

Another international policy framework, the United Nations' 17 SDGs, particularly those targeting poverty reduction and decent work through collaboration and partnership [9], offers a platform for South Africa to address inequality through inclusion. However, their global focus struggles to tackle local systemic barriers, such as the lack of targeted self-employment policies for persons with disabilities. Similarly, the Alma-Ata Declaration of 1978 emphasises health, community participation, and employment for persons with disabilities [15]. Yet, its generic approach fails to address South Africa's unique challenges, like inadequate policy implementation. Additionally, the CBR framework promotes inclusive livelihood initiatives, including meaningful skills training, wage employment, and self-employment, to foster independence and active citizenship [14]. Despite these international inputs, their realisation in South Africa remains limited. These can be achieved through proper administration, accountability and mainstreaming.

In summary, mainstreaming, collaborative work and operating from a single database are necessary. Policy reforms, appropriate attitude, implementation and streamlined systems, as supported by Walt et al. (1994) [6], have the potential to promote self-employment for persons with disabilities, reduce poverty and inequality, and promote inclusive economic growth [3,8,14,15,30,32].

### 4.1. Conclusion, recommendations and implications for practice

The critical findings in this study revealed a complex policy landscape where the absence of self-employment-specific policies for persons with disabilities coexists with potentially leverageable inclusive frameworks. These coexist with potential policy actors who have the potential to facilitate implementation. Leveraging existing policies through effective implementation and targeted policy reforms would ensure the full participation of persons with disabilities in self-employment.

Key policy actors should familiarise themselves with the existing legal framework and emphasise enforcement and consequence management to ensure policies are implemented effectively. Furthermore, a coordinated approach is necessary, involving: a single or integrated system or database to streamline policy implementation and monitoring; a targeted approach that prioritises persons with disabilities in self-employment, and policies that allow for target setting, accurate measurement of targets, and effective monitoring and evaluation. Thus, a policy brief outlining the key findings of this study should be considered, drafted, and submitted to the relevant government department (e.g., Department of Employment and Labour) for further action.

### 4.2. Strengths and limitations

*Strength:* A key strength of this research is the involvement of multiple key policy actors, providing a comprehensive understanding of the complex issues surrounding self-employment for persons with disabilities in a South African context. By incorporating the voices of these policy actors, this research adds valuable knowledge to an under-prioritised and under-researched field. Additionally, this research can be adjusted and replicated in other parts of the world.
*Limitations:* Despite efforts to engage all relevant government departments, not all departments participated in this study, which may limit the generalisability of the findings. This limitation presents an opportunity for future research to build upon this study's foundation and explore the perspectives of additional government departments. Limited literature, specifically on self-employment for persons with disabilities, limited the robustness of this research.

### Acknowledgments

Thank you to all the participants who shared their experiences and the research assistant for her support with data collection. Specifically, as the primary author, I would like to thank Mr. Bareng Mokoena for his insights on the South African legal framework.

### Author contributions

**Conceptualization:** Luther Lebogang Monareng.

**Data curation:** Luther Lebogang Monareng.

**Formal analysis:** Luther Lebogang Monareng.

**Investigation:** Luther Lebogang Monareng.

**Methodology:** Luther Lebogang Monareng.

**Supervision:** Shaheed Mogammad Soeker, Deshini Naidoo.

**Writing – original draft:** Luther Lebogang Monareng.

**Writing – review & editing:** Shaheed Mogammad Soeker, Deshini Naidoo.

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
