## [Decision Letter · Decision Letter 0]

15 Jul 2025

PONE-D-25-26908Examining self-employment policies for persons with disabilities in South Africa: perspectives from policy actorsPLOS ONE

Dear Dr. Monareng,

Thank you for submitting your manuscript to PLOS ONE. After careful consideration, we feel that it has merit but does not fully meet PLOS ONE’s publication criteria as it currently stands. Therefore, we invite you to submit a revised version of the manuscript that addresses the points raised during the review process.

The manuscript is interesting but some reviewers' concerns should be addressed before we can reconsider your paper for publication. 

We look forward to receiving your revised manuscript.

Kind regards,

Davide Costa

Academic Editor

PLOS ONE

Journal Requirements: 

2. In the online submission form, you indicated that [Data can be accessed by contacting the corresponding author].

Reviewers' comments:

Reviewer's Responses to Questions

**Comments to the Author**

1. Is the manuscript technically sound, and do the data support the conclusions?

Reviewer #1: Yes

2. Has the statistical analysis been performed appropriately and rigorously? 

Reviewer #1: N/A

3. Have the authors made all data underlying the findings in their manuscript fully available?

Reviewer #1: Yes

4. Is the manuscript presented in an intelligible fashion and written in standard English?

Reviewer #1: Yes

5. Review Comments to the Author

Reviewer #1: The study presents the results of an original scientific study.

The introduction is too lengthy and sometimes repetitive. It details relevant local international policy frameworks and conventions, but it does not adequately include a critical literature review. For this reason, the study does not appear to be grounded in the existing body of knowledge. Moving ahead, the introduction could be more concisely written. To strengthen, it needs to include more academic literature.

The study meets the applicable ethical standards as the authors obtained ethical approval from an institutional ethics committee.

The following sentence requires correction and revision for clarity: “For instance, the context (environment) or contextual factors such as politics affect the processes followed when formulating content (policies) and outcomes thereafter.” (Lines 81-82). In addition, the word ‘capacitated’ (lines 403-404) reads a bit odd. The authors may want to use a more suitable words such as ‘built capacities’.

The methods section is detailed and indicates that the research was conducted systematically.

The study has generated important and insightful findings, as reported in the findings sections. There are quite a few quotations presented as evidence for each sub-theme. However, the is a limited analysis and interpretation of the findings. Adding more and deeper analysis could strengthen this section. The findings section lacks flow due to the use of too many quotations in the same place without analysis.

Compared to the introduction, the discussion section is stronger. The authors may revise the introduction in line with the discussion that includes many sources, making both the sections consistent.

The reference list needs some revisions and needs to be formatted consistently. For example, the first reference is in all capitals, whereas the rest are not.

6. PLOS authors have the option to publish the peer review history of their article (what does this mean? ). If published, this will include your full peer review and any attached files.

**Do you want your identity to be public for this peer review?** For information about this choice, including consent withdrawal, please see our Privacy Policy .

Reviewer #1: **Yes: ** Md Kamrul Hasan

---

## [Author Response · Author response to Decision Letter 1]

15 Aug 2025

Kindly request that the reviewers provide explicit and constructive feedback, preferably directly in the manuscript.

---

## [Editor Report · Decision Letter 1]

19 Aug 2025

Examining self-employment policies for persons with disabilities in South Africa: perspectives from policy actors

PONE-D-25-26908R1

Dear Dr. Monareng,

We’re pleased to inform you that your manuscript has been judged scientifically suitable for publication and will be formally accepted for publication once it meets all outstanding technical requirements.

Kind regards,

Davide Costa

Academic Editor

PLOS ONE
---

## [Editor Report · Acceptance letter]

PONE-D-25-26908R1

PLOS ONE

Dear Dr. Monareng,

I'm pleased to inform you that your manuscript has been deemed suitable for publication in PLOS ONE. Congratulations! Your manuscript is now being handed over to our production team.

Kind regards,

on behalf of

Dr. Davide Costa

Academic Editor

PLOS ONE